# The Online Processing of Korean Case by Native Korean Speakers and Second Language Learners as Revealed by Eye Movements

**DOI:** 10.3390/brainsci12091230

**Published:** 2022-09-11

**Authors:** Cheryl Frenck-Mestre, Hyeree Choo, Ana Zappa, Julia Herschensohn, Seung-Kyung Kim, Alain Ghio, Sungryung Koh

**Affiliations:** 1Laboratoire Parole et Langage, Aix-Marseille University, 13284 Marseille, France; 2Centre National de Recherche Scientifique, 75794 Paris, France; 3Department of Psychology, Seoul National University, Seoul 08826, Korea; 4Department of Linguistics, University of Washington, Seattle, WA 98195, USA; 5Department of Linguistics, University of Utah, Salt Lake City, UT 84112, USA

**Keywords:** case morphology, Korean, eye movements, L2, reading, visual world

## Abstract

Previous experimental studies have reported clear differences between native speakers and second language (L2) learners as concerns their capacity to extract and exploit morphosyntactic information during online processing. We examined the online processing of nominal case morphology in Korean by native speakers and L2 learners by contrasting canonical (SOV) and scrambled (OSV) structures, across auditory (Experiment 1) and written (Experiment 2) formats. Moreover, we compared different instances of nominal case marking: accusative (NOM-ACC) and dative (NOM-DAT). During auditory processing, Koreans showed incremental processing based on case information, with no effect of scrambling or specific case marking. In contrast, the L2 group showed no evidence of predictive processing and was negatively impacted by scrambling, especially for the accusative. During reading, both Koreans and the L2 group showed a cost of scrambling on first pass reading times, specifically for the dative. Lastly, L2 learners showed better comprehension for scrambled dative than accusative structures across formats. The current set of results show that format, the specific case marking, and word order all affect the online processing of nominal case morphology.

## 1. Introduction

Previous psycholinguistic research has amply documented native speakers’ immediate sensitivity to morphosyntactic factors during online processing, as demonstrated by their ability to use features such as number, gender, or case to predict upcoming elements [1,2,3]. Given this, the role of nominal case marking in canonical and scrambled sentences has been one of extensive investigation in psycholinguistic studies in both auditory and visual contexts and in various European and Asian languages [2,3,4,5,6]. 

In contrast, research on L2 learners has generally revealed a lesser capacity to process morphosyntactic markers online, either to compute structure or to predict upcoming elements [5,7]. Studies that have examined auditory sentence processing have shown dramatic differences between native speakers and L2 learners as concerns their ability to process grammatical case incrementally across several languages, including Korean [8], German [5], and Japanese [6]. During reading, differences between native speakers’ abilities and those of L2 learners are often observed, but to a lesser extent than during auditory processing of utterances [9,10,11], an issue that we address in the present paper. It is unclear whether L2 learners’ shortcomings are to be attributed to an underuse of syntactic information [7,12] or to limited processing capacities, given that several learner and contextual features (e.g., proficiency level, working memory capacity and task demands) influence results for L2 learners [13,14,15]. While both monolingual children and adults of case-marking languages are sensitive to case for predictive processing in non canonical word order (e.g., scrambling), they use a range of top-down and bottom-up cues. In contrast, bilingual children and adults rely more on word order, often misparsing structures in a non-canonical order [8,16,17].

Herein, we present data from two experiments conducted with native speakers and intermediate L2 learners of Korean whose L1 was French, to test the role of case marking and word order during online comprehension in both auditory and written format. We first outline the relevant grammar and orthography of the Korean and French languages and then describe previous research of online processing of case by native and L2 populations. We then develop the present study, which in Experiment 1 used a visual world auditory paradigm and in Experiment 2 adapted the same materials to a reading experiment in which eye movements were recorded. Finally, we discuss our results in terms of the over-arching questions, concluding that the modality of presentation is significant, that the specific type of nominal case plays a role, and that L2 learners show a pattern of processing more similar to that of native speakers during reading than auditory processing. 

### 1.1. Characteristics of Korean

Korean shares numerous phonological and morphological features of languages in the “Altaic” family, which regroups a swath of languages spoken across central and south-east Asia. Among these characteristics are agglutination, head final word order, and vowel harmony. However, these typological similarities have not proven decisive in determining a common ancestry between Korean and the “Altaic” group [18] any more so than between Korean and Japanese [19]. Nonetheless, Korean is indeed a head final, agglutinative language that has canonical SOV word order [20] and presents vowel harmony in a limited class of words in modern Korean [21]. It does not allow for the oral realization of consonant clusters and when they occur in written format only one consonant is produced within a syllable, due to various phonological rules [22,23]. Korean nouns are not marked for gender and only optionally marked for number but are generally marked for case. Examples 1 and 2, below, illustrate the nominative, accusative, dative, and locative case in Korean as well as the topic marker. These case particles are regularly produced, although they may be dropped in informal speech under certain conditions. Korean does not express verbal agreement in either person or number. However, it has a complex morphological system to denote different registers/politeness levels and marks for tense and aspect.

1 오늘은 소년이 학교에서 점심을 먹었다Oneuleun sonyeoni haggyoeseo jeomsimul moegeosstaToday_top_ boy_nom_ school_loc_ lunch_acc_ ate.Today, the boy ate lunch at school. 2 어제 의사가 소년에게 책을 주었습니다Eoje uisaka sonyeonege chaekul jueosseubnidaYesterday doctor_nom_ boy_dat_ book_acc_ gaveYesterday, the doctor gave a book to the boy 

Modern Korean predominately uses the standardized form of Hangul (한글), illustrated in examples 1 and 2, to orthographically represent speech. It is a morphophonemic alphabetic script comprised of 24 letters that represent the 40 phonemes of Korean, which include 19 consonants (14 simple, 5 of which are doubled) and 21 vowels, which comprise 8 simple vowels and a further 13 diphthongs created via the adjunction of either/j/,/w/or/u/. Hangul was created in the mid 15th century [19] to replace the Chinese morpho-syllabic writing system that was originally used to transcribe the Korean language, but which did not sufficiently represent Korean phonemes. The Hangul writing system was adopted over several centuries and originally enjoyed lower social status compared to Chinese. Mixed scripts combining both Chinese and Hangul were the norm up until the 20th century, and Hangul itself was not standardized until the late 20th century [19]. Hangul shows the influence of Chinese in that letters are not written linearly but in syllable blocks (kulja) that generally comprise between 2 and 4 letters (jamo) and obey phonological constraints and spatial properties [24]. Brain imaging studies have examined whether Korean is processed more similarly to Chinese or English, given the combined attributes of being alphabetic and obeying syllable organization [25,26,27]. The orthographic representation of Korean adheres to CV syllable structure such that if a syllable block begins with a vowel, a null grapheme (ㅇ) is inserted prior to the vowel (e.g. 아버지 «abeoji» father), vs. 사과 «sa-kwa» apple). The grapheme to phoneme correspondence in modern Korean is highly regular, if much less so in the early stages of the development of the script [19], i.e., it has a “shallow” orthography with a one-to-one mapping between phonemes and graphemes aside from certain complex codas, but for which consistent phonological rules apply [23]. 

### 1.2. Characteristics of French

French is an Indo-European language, derived from Latin. In contrast to Korean, it is head initial and has canonical SVO word order. Verbs must agree in both number and person with the sentential subject, although for regular verbs the oral realization of agreement is only distinguished for 3 forms across the 6 pronouns; French readers are indeed sensitive to this syncretism [28,29,30]. While French has canonical SVO order for declarative sentences with full noun phrases, it diverges from such when referring to aforementioned nouns, in which case the pronoun is cliticized and rises above the verb, as illustrated in 3a and 3b, below. Clitics are encountered quite frequently, such that native French speakers are well accustomed to processing SOV word order. In contrast to Korean, modern French nouns do not have a morpheme that indicates case for full noun phrases that are direct objects, whether nominative or accusative, however, the dative is marked by the preposition “à” for full nouns, as illustrated in 4a, and by a specific clitic form, as illustrated in 4b. 

3a. Marc voit Jacques.Mark sees Jacques.3b. Marie le voit aussi.Mary him sees also. *Mary also sees him*.4a. Jean parle à JacquesJohn talks to Jack. 4b. Il lui dit des choses importantesHe him says important things. He says important things to him. 

Modern French uses the 26 letters of the Latin alphabet, along with 5 diacritics and two specific forms (“œ” and “ae”), to transcribe speech. It is written linearly and, in comparison to Korean, does not have systematic onsets and codas at the beginning and end of the syllables within a word, although the syllable has been claimed to be more prominent in accessing words in French than say in English [31]. French orthography has a many-to-one mapping as concerns its correspondence with speech; the 36 phonemes of French can be transcribed by 130 different graphemes, which is the cause of considerable difficulty in acquiring the writing system and affects processing written sentences [29,32,33]. A French speaker acquiring Korean may thus find the grapheme to phoneme correspondence far more transparent than in French, which has a deep orthography, but may experience initial difficulties when decoding letters due to their being written in syllable blocks.

### 1.3. Online Processing of Case in Native and Non-Native Speakers

Numerous studies have examined the processing of grammatical case, in the aim of understanding both how it is exploited by native speakers and whether non-native speakers can achieve native-like proficiency. Below, we provide an overview of recent work that is pertinent to this research question. We examine the results from studies of spoken and written presentation successively as the two approaches often address different aspects of processing, and have provided contrasting results for non-native speakers. 

### 1.4. Auditory Processing of Case

An elegant demonstration of how the human parser rapidly exploits grammatical case was provided by Kamide et al. [2] in a visual world experiment conducted in Japanese, which obeys the Head final constraint akin to Korean. Native Japanese speakers listened to auditory sentences while they viewed visual scenes; no additional task was imposed. Of particular interest to the present study were the two constructions that were compared differed in the case marking of the second NP, which was either dative or accusative. Participants’ anticipatory eye movements to the different elements in the scene revealed that for dative utterances (NP1nom-NP2dat) a third NP was projected, in line with the distributional properties of Japanese. This was not the pattern of eye movements observed for accusative utterances (NP1nom-NP2acc), for which a third NP is possible but less likely. Kamide et al. [2] concluded that native Japanese listeners used the combined syntactic information, provided by the case particles on the two first nouns, to predict upcoming arguments. This result was replicated in Japanese with the added factor that the same predictive processing was found for scrambled dative structures [6]. The same conclusion was reached in two visual world paradigm studies conducted in German [3,5]. Utterances had either canonical SVO or scrambled OVS word order. Independent of word order, native speakers predicted the correct thematic role for the second noun, based on the case marked determiner preceding the first noun and semantic restrictions of the verb, and directed their gaze to it prior to its enunciation. Akin to the data from Japanese [2,6], no differences were found between canonical (SVO) and scrambled (OVS) word order until after the information from the first two words had been integrated.

Recently, we used a modified visual world paradigm to investigate this question in Korean [8]. Participants saw two line drawings for 1 s, as illustrated in Figure 1, prior to the onset of the utterance that described one of the two drawings. The utterances had either canonical SOV or scrambled OSV word order for both monotransitive accusative and dative structures and all nouns were marked for case. In monoclausal sentences, scrambling does not require long-distance movement involving filler-gap configurations but does require the processing of case-marking that indicates grammatical roles [34]. The eye movement record showed unequivocally that native speakers exploited case to predict the structure of the utterances. No effects were found during the auditory processing of the first noun. However, starting at the second noun, participants directed their gaze to the correct image based on the combined case marking of the two nouns. This was true irrespective of word order (SOV vs. OSV) and structure (accusative or dative). Hence, native Koreans showed no particular cost of scrambling in syntactically simple utterances (cf. [6], in Japanese) and rapidly integrated case marking to predict the final interpretation.

The results obtained for native speakers contrast sharply with those reported for L2 learners. In 3 independent studies, conducted in Korean [8], Japanese [6], and German [5] that used the same paradigm as the studies conducted with native speakers [2,3,8], L2 learners failed to exploit case online. No anticipatory looks to the correct element in the scene were observed when processing either nominal case particles [6,8] or case marked NPs and verb semantics [5]. Furthermore, L2 learners responded distinctly to dative as opposed to accusative particles [8]. When L2 learners did make anticipatory looks, they systematically adopted a strategy based on canonical word order, thus ignoring the information provided by case in scrambled utterances [5,6]. Across these studies, the common pattern of results was either the delayed integration of case information, or even the absence of integration. Moreover, this was found for advanced late learners as well as for less proficient learners [5] and for L2 learners whose L1 was either typologically similar or distant as concerns the system of nominal case particles [8]. 

Native speakers and L2 learners thus differ substantially in their ability to exploit case during auditory processing. However, one can question whether the transitory nature of speech played an important role. Indeed, in all three L2 studies cited above, the learners did eventually converge upon the correct interpretation (but see [8]). The question, thus, is whether written format would provide a different pattern of results, if not as concerns predictive processing but the ability to immediately exploit case information altogether. Several studies, summarized below, have addressed this question.

### 1.5. Written Processing of Case

The majority of studies of written processing of case in L2 learners have examined single sentences and have been conducted within the framework of structural-based accounts of parsing [11,35]. Several studies have been conducted in German, due to its robust case marking system, which presents a particular challenge for adult L2 learners [9,10,11,14,15,35,36]. These studies have often exploited structural ambiguities, which can arise due to syncretism in German case morphology. The results provide a complex answer to the question of whether and when adults who have acquired a second language after early childhood incorporate the grammatical/thematic information provided by nominal case to process sentence meaning. Hopp [9] and Jackson [10] reported that “near-native” and highly proficient L2 participants performed in a similar fashion to native controls. Both groups showed increased self-paced reading times at the critical NP for scrambled (OS) compared to canonical (SO) word order in embedded relative clauses. In contrast, intermediate L2 learners either did not show online effects or only at sentence end. However, in a replication study that recorded eye-movements, Jackson and colleagues [11] concluded that intermediate L2 readers do in fact show immediate sensitivity to grammatical information provided by case marked determiners. Two other self-paced reading studies conducted in German provided mixed results. One showed that advanced L2 learners displayed online sensitivity to case and a pattern of processing similar to, but not identical to, native speakers [15], while the other, despite being a close replication, did not [14]. However, the two studies differed in task demands as concerns the requirement to make a grammatical judgment, which may have prompted participants to make explicit use of case.

Despite the numerous factors that apparently come into play in L2 processing, the overall picture from the above cited studies suggests that, even if not part of the L1 grammar, grammatical case can be exploited online during reading once the learner has achieved sufficient proficiency. This differs from the overall consensus for auditory processing, however; none of the studies that examined auditory processing recruited “near native” or even highly proficient learners.

### 1.6. Present Study

Herein, we examined the online processing of nominal case particles in Korean. As an overt case-marked language with canonical SOV constituent word order that allows for movement (scrambling) of arguments, Korean provides a rich test bed to explore processing of case. We compared processing for native Koreans to that of L2 learners of Korean whose native language was French, which is not overtly case-marked, shows SVO constituent order and disallows scrambling. To address the question of how the modality of presentation may affect the capacity to process case online, we compared auditory processing (Experiment 1) to reading (Experiment 2). We examined processing for both canonical (SOV) and scrambled (OSV) word order. 

To delve deeper into the question of Korean case, we compared processing for dative and accusative structures. There are numerous differences between the behavior of two “grammatical” case particles, the nominative and accusative, and “semantic” ones, notably the dative [37]. One such difference concerns ellipsis, i.e., where the particle is dropped. Various offline studies of Korean have shown that the nominative and accusative are subject to ellipsis [38,39,40,41], although ellipsis obeys constraints [19,20,39,42,43,44,45]. In contrast, ellipsis is rarely attested for dative case, which can be attributed to its ranking [43,46]. Dative case particles also differ from the nominative and accusative as concerns the alternation of phonological form. To maintain CV structure, the majority of nominal case particles are phonologically constrained, having +V structure for nouns that have a coda consonant (VC# or CVC#) versus +CV for nouns that do not (V# or CV#). To illustrate, there are two forms for the nominative (이/가 i/ka) accusative (을/를 ul/lul), instrumental (으로/로 ulo/lo), and vocative (아/야 a/ya), as well as for the topic particle (은/는 un/nun), suffixed to nouns ending with a consonant and vowel, respectively. These variations are purely phonological, with no semantic content. This stands in contrast to the dative case marker, which also has two forms (에/에게 ey/eykey), but which alternates based on the semantic properties of the noun it attaches to (inanimate and animate, respectively; idem for the locative (에서/에게서 eyseo/eykeyseo). Studies of L2 learners and heritage speakers have shown that the various phonetic, grammatical, and semantic differences for the dative as opposed to the nominative and accusative particles affect the ability to produce and understand these particles, with a definite advantage for dative [8,47].

In relation to scrambling in Korean, not all orders are permissible, depending on whether two NPs share the same grammatical case and/or animacy value, whether they are both arguments, etc. [48,49,50]. In the present study, we considered rather simple instances of permutations, contrasting canonical SOV and scrambled OSV orders for both monotransitive accusative and dative structures, as illustrated in 5a through 6b. All materials had the same structure as in these examples, with two initial NPs marked as either nominative and accusative or nominative and dative, in either canonical or scrambled word order, followed by the verb.

5a. 아이가 어머니를 깨운다.aika eomeonilul kkaeuntachild_nom_ mother_acc_ wake-Pres Ind‘The child wakes the mother’5b. 어머니를 아이가 깨운다.Eomeonilul aika kkaeuntamother_acc_ child_nom_ wake-Pres Ind‘The child wakes the mother’6a. 환자가 의사에게 이야기한다Hwanjaka uisaeykey iyakihantapatient_nom_ doctor_dat_ say hello to-Pres Ind‘The patient says hello to the doctor’6b. 의사에게 환자가 이야기한다Uisaeykey hwanjaka iyakihantadoctor_dat_ patient_nom_ say hello to-Pres Ind‘The patient says hello to the doctor’

For the structures illustrated above, both orders are permissible although SOV order is canonical and observed more frequently than scrambled OSV order [36,49]. Frequency of occurrence may play a greater role in processing for non-native than native speakers [5,6,8]. There is also debate concerning the processing cost linked to scrambling [36,49,51,52,53]; for a more in-depth discussion, we refer the reader to Frenck-Mestre et al. [8]. Non-canonical OSV scrambled structures may be considered syntactically more complex (cf. Frenck-Mestre et al. [8] for further discussion), but whether this is evidenced in processing depends upon the method and structures employed [52,53,54,55,56,57,58]. If participants rapidly parse scrambled elements, no immediate processing cost may appear [45,56,57,59]. 

Given earlier research, we predict that native speakers will use case and word order as cues during online processing for both auditory and written sentences. This should translate, during auditory processing, into a greater proportion of looks to the correct image, starting from the onset of the second noun, due to native speakers predicting the upcoming elements of the sentence based on the case marking of N1. They should exploit case information incrementally as the utterance unfolds to build a syntactic representation, whether for canonical or scrambled word order. During reading, if there is a cost associated with scrambling, native readers should show increased reading times at the first noun, for scrambled (OSV) sentences compared to canonical (SOV) word order. For L2 learners, we predict that they will be insensitive to case during auditory processing, relying on word order, in line with previous results [5,6,8]. Thus, no preference for either image should be apparent prior to the processing of the verb and our L2 learners should misinterpret scrambled (OSV) utterances. However, we expect to observe an advantage in the L2 learners for dative as opposed to accusative case [8]. During reading, which may allow L2 learners more time to process the various elements given that we used eye tracking and no time constraints were imposed, we hypothesize that L2 learners may show more native-like processing. Hence, if scrambled structures incur a cost, we predict increased reading times at the initial noun compared to canonical structures in a similar fashion to native readers of Korean. 

The current study presents several advantages. First, we examined online processing across formats for the same linguistic materials. Second, the recording of eye movements provides a millisecond precise record of processing, which allowed us to pinpoint when participants exploited case during auditory processing and a finer grained measure than self-paced reading to determine whether scrambling incurred a cost. Third, the direct comparison of different types of case marking is novel. Last, we address the currently debated “replication crisis” in that our first experiment employed the same paradigm and materials as Frenck-Mestre et al. [8], but with new participant groups.

## 2. Experiment 1

The first experiment served both as a basis of comparison for our second experiment and an extension of our earlier work (Frenck-Mestre et al. [8]). In that study, we performed a cross-linguistic comparison by examining the performance of two L2 learner groups whose native language was either typologically close to Korean as concerns nominal case marking and head final properties (Kazakh) or typologically distant (French) to a group of native Koreans. As outlined above, the typological distance between the learners’ L1 and Korean did not suffice to explain the pattern of results. In relation to our previous study, the L2 learner group recruited for the present study comprised what can be termed as “classroom learners”, having acquired Korean in a university curriculum and living in their country of origin (France) without any outside use of Korean. None had lived for any length of time in Korea or a Korean-speaking community in France. This factor is of importance, given that our previous results were reported for L2 learners living in Korea, who had been immersed in the Korean language for several months. As such, it is possible that the current study may not replicate the previous pattern of results as concerns the differential sensitivity to dative and accusative structures, if such stems from exposure as opposed to any explicit instruction. Indeed, the distributional properties of a language are implicitly acquired via experience.

We used the modified visual world paradigm, illustrated in Figure 1, above, to follow participants’ online processing of auditory sentences. The auditory materials were composed of simple declarative sentences such as illustrated in examples 5 through 6, in which we manipulated word order (SOV vs. OSV) and sentence structure (accusative monotransitive vs. dative). All nouns were marked for case, as is appropriate for sentences devoid of context in Korean [59].

### 2.1. Methods

#### 2.1.1. Participants

Twenty-seven native speakers of French (25 female, *M*_age_ = 21.4 years, range = 18–27, SD = 2.2), enrolled in their 3rd year of Korean at Aix-Marseille University (AMU) (*M* = 3.7 years, range = 6–7 semesters, SD = 0.4 mo.) and 16 native Korean speakers (10 female, *M*_age_ = 18.7 years, range = 18–20, SD = 8 mo.) enrolled in their first or second year at Seoul National University (SNU) participated. None had participated in our previous study [8]. None presented ocular-motor deficits or history of neurological insult. All participants gave informed written consent and were monetarily compensated. The study was approved by the ethics committees at AMU (IRB approval for AAPRI-4-2016) and SNU (IRB approval N° 1604/003-001).

L2 participants completed a language background questionnaire that queried their duration of stay in Korea (13 participants had never been to Korea or immersed; the mean for the other 14 was 2.6 mo., range = 1–6 mo., SD = 1.9 mo), first exposure to Korean (*M* = 18.6 years, range = 16–23, SD = 1.9 years) and aspects of self-rated proficiency. In a post-test measure of vocabulary and case morphology, all participants correctly produced the written French translations of the materials and gave the correct grammatical description of the nominal case markers (*M*_score_ = 97.3%, range = 94–100%, SD = 2.1%)), as did the Korean controls (*M*_score_ = 100%, SD = 0%).

#### 2.1.2. Stimuli

Forty auditory sentences, each paired with a visual scene comprised of two line drawings, served as experimental stimuli (see Appendix A). The materials were taken from our previous study [8]. We used a within participant 2 × 2 design, defined by Case (dative vs. accusative) and Order (canonical (SOV) vs. scrambled (OSV)). Each auditory sentence was presented in canonical and scrambled word order across 2 counter-balanced lists. Each list contained 20 accusative and 20 dative utterances, with 10 of each type in canonical and in scrambled order. The sentences were created from a set of 5 monotransitive accusative and 5 dative verbs and 20 pairs of concrete nouns. All nouns were marked for case (nominative, accusative, dative). All verbs were produced in the inflected form for standard newspaper reporting. Each of the 10 verbs were presented in 4 auditory sentences with 4 different noun pairs. All nouns were marked as object (dative or accusative) and as subject equally often across the set of 5 dative and 5 accusative verbs. For 3 of the 5 dative verbs a third, accusative noun was presented. This was due to our selection of high frequency verbs and their constraints. For all dative utterances, the accusative noun was always in the third position. All nouns and verbs were selected to be part of the L2 learners’ vocabulary. The pairing of two nouns with a given verb was constrained by semantics and by the possibility of creating unambiguous visual scenes. Twenty filler sentences comprising various structures with one or two nouns and a verb were also presented in each list. (e.g., *dog**_nom_ cat**_acc_ bites*; and *paper**_nom_ white is*).

Utterances were recorded by a linguistically trained female native Korean speaker, using neutral intonation, in a sound attenuated recording studio at 48kHz (32-bit float). They were roughly 2 s long. The onset and duration of each word was determined post-recording using SPPAS (www.sppas.org, accessed on 15 September 2017) and verified with PRAAT (mean durations in msec for N1, N2 and VB in accusative utterances were 723, 710, and 636 for canonical and 692, 693, and 618 for scrambled order, and for datives were 703, 882, and 712 for canonical and 845, 717, and 695 for scrambled order. The comparison of the durations for N1 and N2 across Case and Order revealed that accusatives were overall shorter in duration than datives (β = −39.49, se = 8. 06, t = −4.90, *p* < 0.001, but there was no effect of Order (β = 9.44, se = 8. 06, t = 1.17, *p* < 0.24) or the interaction (β = 3.64, se = 8. 06, t = 0.45, *p* < 0.65). These onsets were used as inaudible triggers sent during the eye movement recording to compute the location of the participants’ gaze. 

A professional artist created the line drawings using India ink and paper, which were subsequently digitized on a template comprised of 2 equally sized rectangles, at a resolution of 1024 × 768 pixels. The drawings are available upon request. Each drawing consisted of 2 complementary scenes, depicting either the first or the second noun as the subject (see Figure 1). The correct image appeared equally often on the left or right across trials and experimental conditions. All drawings were presented only once, in conjunction with a unique utterance. 

#### 2.1.3. Procedure

Participants sat 60 cm from a CRT screen in a dimly lit room with their head restrained by a chin and forehead rest. They were asked to listen to sentences and select the image depicting it via a response box. Participants’ eye movements were recorded from the right eye using either Eyelink Tower mount 1000 (sampled at a rate of 500 hz) at AMU or Eyelink II head mount (sample rate of 250 hz) at SNU https://www.sr-research.com, accessed on 15 August 2022). The difference between the two systems was thus the sample rate, being either 2 or 4 msec; all else was equal across set-ups and the 2 msec difference between systems is negligible in terms of gaze durations and the programming of saccades. Eye movements were calibrated at the outset of the experiment using a nine point calibration grid. Drift checks were performed at the outset of each trial and recalibrations were performed as necessary. Each trial began with a central fixation point (500 ms) and warning tone followed by the visual scene. The auditory sentence was presented 1000 ms after the onset of the visual scene, which remained on the screen until the participants’ response. The next trial began immediately thereafter, and no feedback was provided. The entire session lasted roughly 30 min.

### 2.2. Results

#### 2.2.1. Statistical Analysis

We used generalized mixed effects models to analyze both accuracy and dwell time [60]. Dwell times were calculated from the auditory onset of a given region of interest (ROI) to the onset of the next ROI for the first two ROI (N1 and N2) and from the onset of the final ROI (VB) to the onset of the manual response for the verb. For the dative sentences that contained a third (accusative) noun prior to the verb, this time was not included in the analyses. For dwell times, a binary variable was constructed based on the amount of time spent on each of the two images. Hence, on a given trial, if a participant spent 220 msec on the correct image during the processing of a given auditory word and 130 msec on the incorrect image, then 220 “1” values and 130 “0” values were entered into the logistic regression model. This ensured that the variability in dwell times across trials and participants was retained in the model while respecting the binary nature of the data. We report the results from the maximal models [61] provided that the models converge.

#### 2.2.2. Accuracy

The first model compared the 2 groups and included the sum-coded factors Group (Korean native vs. L2 learners), Case (Accusative vs. Dative), and Order (Canonical vs. Scrambled), and their interactions. Participant was included as a random factor, item was not included, and no slope was included due to non-convergence. The results are summarized in Table 1.

Accuracy was lower overall for L2 learners (64%) than for Korean native controls (96%) (β = −1.40, se = 0.15, z = −9.03, *p* < 0.001). The effect of Group was modified by interactions with Order (β = 0.54, se = 0.12, z = 4.37, *p* < 0.001), and Case (β = −0.34, se = 0.12, z = −2.76, *p* < 0.006), and the higher order interaction involving Case and Order (β = 0.31, se = 0.12, z = 2.47, *p* < 0.01). Independent models were run on the data for each group, using the same model structure sans the fixed factor Group. 

Korean native speakers had above chance accuracy (β = 3.39, se = 0.29, z = 11.90, *p* < 0.00001) and showed an effect of Case (β = 0.48, se = 0.23, z = 2.09, *p* < 0.05) due to slightly higher accuracy for accusative than dative utterances (98% vs. 94%, respectively). There was no effect of Order (β = −0.14, se = 0.23, z = −0.61, ns), nor was Case modified by Order (β = −0.01, se = 0.23, z = −0.04, ns).

L2 learners also had above chance accuracy (β = 0.73, se = 0.13, z = 5.68, *p* < 0.001) and main effects of Case (β = −0.19, se = 0.08, z = −2.43, *p* < 0.01) and Order ((β = 0.91, se = 0.08, z = 11.52, *p* < 0.001) as well as their interaction (β = 0.57, se = 0.08, z = 7.25, *p* < 0.001). Accuracy was higher for canonical than scrambled word order, however, the effect was greater for accusative (β = 3.14, se = 0. 30, z = −10.33, *p* < 0.001) than dative utterances (β = −0.77, se = 0.28, z = −2.71, *p* < 0.01).

#### 2.2.3. Dwell Times

Independent analyses were performed on dwell times recorded during N1, N2, and the Verb on correct trials. The first model included the sum-coded fixed factors Group, Case, Order, and their interactions. Participant and Item both included random intercepts and Participant included random slopes for Case and Order. This model also had the lowest AIC value. 

The percentage of dwell times are presented in Table 2 as a function of the image fixated (Correct vs. Incorrect) during each auditory word (N1, N2, and Vb), for each Case (Accusative vs. Dative) Order (Canonical vs. Scrambled) and Group (Korean vs. L2), for both correct trials and all trials. To illustrate the data pattern, we present density plots for Korean participants in Figure 2 and for L2 learners in Figure 3a,b. The plots clearly demonstrate that for the duration of N1, participants directed their gaze almost exclusively to the correct or incorrect image without regard to case marking. During N2, participants again directed their gaze almost exclusively to one or the other image, but while L2 learners were at chance, Korean native speakers looked more often at the correct image. During the verb, Koreans concentrated their gaze on the correct image, while for L2 learners this was dependent on both Case and Order. The mirror image of looks toward the correct and incorrect image reveals that participants did not direct their gaze outside of the two images.

At N1, the interaction involving Group and Case was significant (β = −0.11, se = 0.06, z = −1.90, *p* < 0.05), as was the higher order interaction involving Group, Case, and Order (β = 0.09, se = 0.003, z = 25.30, *p* < 0.001). At N2, the main effect of Group was significant (β = −0.13, se = 0.05, z = −2.64, *p* < 0.01), as was the interaction between Group, Case, and Order (β = 0.03, se = 0.002, z = 12.01, *p* < 0.001). At the Verb, the same pattern emerged, with a main effect of Group (β = −0.17, se = 0.06, z = −2.85, *p* < 0.004), and the higher order interaction involving Group, Case, and Order (β = 0.02, se = 0.002, z = 10.92, *p* < 0.001). Subsequent models were performed on the data for the two groups independently. For both groups and for all ROI, the model included the sum coded fixed factors Case and Order and their interaction, with a random intercept for Participant and Item and random slopes of both Case and Order for Participant.

For Koreans, no effects emerged at N1 (Intercept (β = −0.03, se = 0.11, z = −0.23, ns); Case (β = −0.16, se = 0.12, z = 1.27, ns); Order (β = −0.10, se = 0.12, z = 0.83, ns); Case and Order (β = −0.15, se = 0.10, z = −1.54, ns). The percentage of dwell time did not differ for the correct and incorrect image at this point in the utterance, independent of the case marking or word order. Starting at N2, the percentage of dwell times was significantly higher for the correct than incorrect image (β = 0.42, se = 0.14, z = 2.98, *p* < 0.003), with no effect of Case (β = −0.08, se = 0.13, z = 0.61, ns) Order (β = −0.16, se = 0.15, z = −0.11, ns) or their interaction (β = −0.06, se = 0.19, z = −0.67, ns). Native Koreans looked at the correct image more often than the incorrect image when listening to the second auditory noun, irrespective of Case or word Order. At the final region, the Verb, the same pattern was observed, whereby significantly more looks were directed to the correct image (β = 1.09, se = 0.18, z = 6.22, *p* < 0.001) with no other effects (Case (β = −0.04, se = 0.12, z = 0.43, ns), Order (β = −0.05, se = 0.12, z = −0.37, ns), Case × Order (β = 0.11, se = 0.10, z = 1.03, ns).

Independent of Case or Order, Korean participants looked at the correct image the vast majority of the time during the processing of the verb and until their response. These results are summarized in Figure 2.

For L2 learners, no effects were observed at the first noun: Intercept (β = −0.18, se = 0.18, z = −0.99, ns), Case (β = −0.08, se = 0.17, z = −0.48, ns), Order (β = 0.25, se = 0.18, z = 0.37, ns), Case × Order (β = 0.06, se = 0.15, z = 0.37, ns). At the second noun, again, no effects were significant: Intercept (β = 0.28, se = 0.18, z =−1.50, *p* < 0.13), Case (β = 0.16, se = 0.19, z = 0.85, ns), Order (β = −0.11, se = 0.19, z = −0.59, ns), Case × Order (β = −0.10, se = 0.17, z = −0.58, ns). These results are depicted in Figure 3a At the final Verb, there were effects of Intercept (β = 0.62, se = 0.07, z = 9.06, *p* < 0.00001) and Case (β = −0.15, se = 0.06, z = −2.46, *p* < 0.01). The effect of Order did not reach significance (β = 0.02, se = 0.06, z = 0.43, ns) and the interaction term was marginal (β = 0.09, se = 0.05, z = 1.89, *p* < 0.06). For both accusative and dative utterances, L2 learners showed significantly greater dwell times on the correct image for datives when processing the verb and up to their response. These results are depicted in Figure 3b.

Due to the high error rate for scrambled word order for the L2 learner group, we conducted a further set of analyses that included all trials for all conditions using the same models as performed on the data for correct trials alone. This was to ensure that the pattern found for the correct trials was only representative of the entire data set. The results showed a very similar pattern across the two data sets for the first two nouns, but a dissimilar pattern at the verb. At N1, no effects were significant (Intercept (β = −0.003, se = 0.09, z = −0.03, ns), Case (β = −0.03, se = 0.09, z = −0.29, ns), Order (β = 0.13, se = 0.09, z = 1.53, ns), Case × Order (β = −0.004, se = 0.07, z = −0.05, ns)). The same was true at N2 (Intercept (β = 0.12, se = 0.08, z = 1.55, *p* < 0.12), Case (β = 0.004, se = 0.09, z = 0.05, ns), Order (β = 0.07, se = 0.09, z = 0.81, ns), Case × Order (β = −0.01, se = 0.06, z = −0.15, ns). In contrast, at the final verb, there were significant effects of the Intercept (β = 0.24, se = 0.06, z = 4.29, *p* < 0.001), of Case (β = −0.14, se = 0.06, z = −2.34, *p* < 0.01), Order (β = 0.17, se = 0.05, z = 3.19, *p* < 0.001), and the interaction of Case and Order (β = 0.12, se = 0.04, z = 2.75, *p* < 0.01). L2 learners showed significantly higher dwell times on the correct than incorrect image, however; different patterns emerged for accusative and dative sentences. For accusatives, L2 learners spent more time looking at the correct than incorrect image for canonical (63%) but not for scrambled word order (44%). For dative utterances, L2 learners showed greater dwell time on the correct than incorrect image for both canonical (63%) and scrambled word order (60%). These results are depicted in Figure 3b. 

### 2.3. Discussion

Our eye movement data showed that native Koreans rapidly exploited case marking during auditory sentence processing and performed at ceiling level. From the onset of the second auditory noun, Korean participants directed their gaze significantly more often to the image that correctly depicted the utterance (roughly 60% of gaze duration) and the effect increased as the utterance unfolded, with roughly 75% of dwell time on the correct image during the processing of the verb. This was true independent of case or word order. In addition, they showed ceiling level accuracy. This pattern corroborates results from online studies of case in other languages [5,6] and replicates results in Korean [8]. The ease of exploiting nominal case also replicates the results from numerous offline studies conducted with native Koreans across various tasks [38,47,59]. 

The eye movement data for the L2 group provides no evidence of the use of case to predict the structure of the utterances. Indeed, L2 learners showed no preference for the correct image prior to the final verb, even for canonical word order. While there was a numerical difference in favor of the correct image at N2, the effect was not reliable. Interestingly, the L2 learners showed different patterns for the accusative and dative. For datives, they showed a small but reliable increase in looks to the correct image at the verb independent of word order. For accusatives, they showed increased dwell times on the correct image for canonical word order, but on the incorrect image for scrambled word order. This interaction effect was not reliable when only correct trials were taken into account, but very robust when all trials were considered. Given that roughly 70% of trials were associated with the incorrect choice of the image for scrambled accusatives, we consider the analyses of all trials a better indicator of performance for the eye movement data. The eye movement pattern we observed for all trials indeed mimics the pattern of accuracy data. The specific difficulty with scrambled word order for accusative utterances may be linked to various factors [8,47,62]. Among these, the phonetic salience of case markers may play a role; indeed the dative marker for animate nouns “에게” (“eykey”) is bisyllabic, whereas the accusative marker is not only monosyllabic but subject to variation depending on phonological constraints (*을“ul”* and 를 “*lul”)*. If the L2 learners were predominantly using word order rather than case marking to compute sentence meaning, as both the accuracy and eye movement data suggest, the absence of a systematic and salient phonetic realization of the case marker may have further impaired their use of case. 

Given the present results in conjunction with those of our previous study (Frenck-Mestre et al. [8]), it would be tempting to conclude that we have provided unequivocal evidence that, at least in the beginning stages, French learners of Korean use word order preferentially over case, even when all nouns are clearly case marked, and that the accusative shows a particular disadvantage in relation to the dative. However, prior to reaching said conclusion, it is worthwhile to examine the conditions under which our L2 learners computed case markers online. Notably, as in numerous L2 studies of the online processing of case, participants were confronted with auditory materials [4,5,6,47]. Hopp [4] clearly demonstrated that even native speakers fail to use case, choosing word order instead, when auditory materials are difficult to process (e.g., under speeded conditions or in noise), and argued that what is lacking in L2 performance is just that, performance limitations due to proficiency, processing speed, etc., rather than a true representational deficit that would distinguish them from native speakers. While the present auditory materials were presented at normal speech rate, and indeed accuracy for canonical word order was high, it is possible that the transient nature of speech may at least partially explain the poor performance of the L2 group for scrambled utterances. The effect of the mode of presentation was investigated in our second experiment, in which we created a written version of materials, tested with a new group of native controls and L2 participants.

## 3. Experiment 2

In this experiment, we created a written version of our materials to reexamine the online use of case versus word order for dative and monotransitive accusative sentences via the recording of eye movements during reading. Two new groups of participants were recruited, comprising a group of native Korean speakers and an L2 learner group whose native language was French. Both groups were extracted from the same populations, as in the first experiment, to ensure homogeneity across the two experiments. 

As outlined in the general introduction, numerous studies comparing native and L2 participants’ ability to use morphosyntactic information, and notably case marking, during reading have provided evidence that while native speakers almost systematically outperform L2 readers, performance in the latter group varies as a function of proficiency [4,9,10,14,35], the availability of information structure [35], working memory capacity [13], task demands [14,15], and the ability to re-read elements [11]. We thus expected performance to vary as a function of group, with superior online interpretation of case particles to compute sentence meaning in the native control group compared to the L2 group we recruited, which could be considered low-intermediate with no immersion experience. Nonetheless, in comparison to the results for L2 learners from Experiment 1, showing a dominance of word order over case marking, especially for accusative sentences, we predicted improved performance during reading in line with the studies cited above and in the general introduction. 

In relation to the effect of scrambling, the present experiment also allowed us to test the hypothesis that such may incur a cost [55,57,58,63]. In a previous self-paced reading study that compared the processing of canonical versus scrambled Japanese sentences, no effects were found on reading times [55]. Nonetheless, given that we used eye tracking to measure sentence processing during reading of entire sentences rather than present sentences in segments or word by word, as is the case for self-paced reading, the pattern of results may differ from previous work. The recording of eye movements not only allows a finer grain of investigation than self-paced reading, but allows participants to re-read the sentence or parts thereof at will due to sentences being presented in full. As such, we may find greater sensitivity to scrambling than reported in studies using self-paced reading. A similar argument was forwarded as concerns the ability to exploit case [11]. Intermediate learners of German apparently showed greater sensitivity to case marked NPs in the eye-movement record [11] than in self-paced reading [10]. Nonetheless, said claim should be accepted with caution due to the fact that in the eye movement study the 3-letter determiner was defined as a critical region and was examined independently of the subsequent noun or previous pronoun. There is a wealth of research from the field of reading suggesting that this is far from optimal, given the short length and the grammatical function of determiners [64]. 

As regards the extraction of case information during reading in Korean, there is some evidence from native speakers that they process nominal case particles in parafoveal vision, as demonstrated in a boundary paradigm [65]. Similarly, this implies that Korean readers can extract syntactic/semantic information from letters outside foveal vision (cf. Wang et al. [66], for additional evidence), a result that has been the subject of considerable debate in languages that use the Latin alphabet [67]. In the present experiment, the length of nouns varied from 2 to 3 syllable blocks and varied in the number of letters within the different syllable blocks (e.g. 남자 namja vs. 아이 ai, 요리사 yorisa vs. 선생님 seonsaengnim). The case particles were represented by a single syllable block for the nominative (이/가 i/ka) and accusative (을/를 ul/lul) and two syllable blocks for the dative (에게 eykey). Hence, the nouns plus case particles varied in length from 3 to 5 syllable blocks, with varying complexity. Choi and Koh [68] reported that Korean’s perceptual span extends from the point of fixation to roughly 7 syllable blocks to the right (i.e., in the direction of reading) and 1 to the left. Hence, for the vast majority of the nouns comprised in the current study, the case particle could be processed as part and parcel of the noun. We therefore did not attempt to specifically isolate the case particles in analyses. Moreover, as stated above, extremely short regions are not the candidate of choice. 

We predicted that native Korean readers would show ceiling level performance as concerns comprehension and that such should not differ as a function of either word order (canonical SOV vs. scrambled OSV) or structure (monotransitive accusative vs. dative). For reading times, if scrambling incurs reordering, we should find elevated gaze durations during the initial encounter of scrambled structures (NP1acc–NP2nom and NP1dat–NP2nom); whether this would vary as a function of structure is an open question. For L2 learners of Korean we predicted higher comprehension for canonical than scrambled sentences, although in comparison to the auditory presentation we predicted a lesser difference between the two conditions due to the possibility to either dwell on the nouns or reread them. For reading times, we predicted a significant cost for scrambled sentences on the first 2 nouns, which we examined independently for accusative and dative structures due to inherent length differences. 

### 3.1. Methods

#### 3.1.1. Participants

Sixteen native French speakers (*M*_age_ = 21.6 years, range = 19–24, SD = 1.4) enrolled in the third year of Korean studies at Aix-Marseille University (AMU) (*M* years of study of Korean = 3.3, range = 3–4 years, SD = 5 mo.) and 16 native Korean speakers (*M*_age_ = 18.7 years, range = 19–20, SD = 8 mo.) enrolled in their first or second year at Seoul National University (SNU) participated. None had taken part in Experiment 1. All L2 participants had achieved reading and writing fluency in Korean according to their academic record at AMU. All participants gave informed written consent prior to the study and were monetarily compensated; none presented any ocular-motor deficits or history of neurological insult. The study was approved by the ethics committees at AMU (IRB approval for AAPRI-4-2016) and SNU (IRB approval 1604/003-001).

L2 participants filled out the same language background questionnaire and measure of vocabulary in Korean and case morphology as in Experiment 1. They were all able to produce the written French translations of all Korean materials as well as give the correct grammatical description of the case markers (*M*_score_ = 97.5%, range = 92–100%, SD = 3%), as were the Korean controls (*M*_score_ = 100%, SD = 0). They had not lived in Korea or a Korean speaking community for more than 2 months (9 participants had never been to Korea or immersed; the mean for the other 7 was 1.6 mo., SD = 6 mo.) and had not been exposed to Korean before age 18 years (*M* = 19.1 years, range = 19–21, SD = 1.3 mo.). 

#### 3.1.2. Stimuli

The same 40 experimental sentences and 20 filler sentences used in Experiment 1 were presented, but in written Korean as opposed to auditory format (see Appendix A). All sentences were followed by a probe question that required a binary (yes/no) response. Probe questions began with a noun bearing the topic case marker, followed by the same verb as in the sentence and a question mark. The topic marked noun in the probe sentence was either the subject marked noun (50% of trials) or the object marked noun (50% of trials) in the preceding sentence, hence half of the probe questions required a positive and half a negative response. Two counter-balanced lists were created such that all experimental sentences were seen in both canonical and scrambled word order but in only one condition per list. Sentences were presented in a semi-random order, with the restriction that no more than 3 experimental sentences follow each other. 

#### 3.1.3. Procedure

This was identical to that used in Experiment 1 as concerns the equipment, the recording of eye movements, and calibration procedure. All materials were presented in written Korean (Hangul). Sentences varied in length from 3 to 4 words and were presented on a single line. Accuracy of measurement was maintained at less than one character via the initial calibration and subsequent validation checks throughout the experiment. A trial began with a warning tone followed by a fixation point placed at the left edge of the CRT screen for 250 ms, which was replaced by the stimulus sentence, starting in the same position as the fixation point. Participants were instructed to fixate the fixation point, read the sentence silently and respond via the button box when they had understood the sentence. A probe question immediately followed, to which participants made a binary response on the button box. Participants were provided with 3 short breaks during the recording session and were allowed to request a break if needed. The experimental session lasted roughly 20 min, followed by participants’ completion of the questionnaires. 

### 3.2. Results

#### 3.2.1. Statistical Analysis

We used R packages lme4 [60] (glmer) to analyze accuracy data and linear mixed effects models (lme) to analyze reading times on the different regions of interest (ROI). We report the results from the maximal models [61] provided that the models converge. For reading times, first pass measures included first fixations and first pass dwell times, i.e., all fixations in an ROI that originated from the left of the region and prior to exiting it. Later measures included total dwell times, i.e., all fixations in an ROI from onset of the sentence until the end of the trial, and regressions, i.e., whether or not participants made a regressive saccade to N1 or N2. Second pass dwell times were not considered due to the inconsistent nature of this measure; indeed, when participants do not re-read a region, this measure renders a null value, as was true for 46% of native speakers’ data. Total dwell times are thus the preferred measure of later effects during reading. 

#### 3.2.2. Accuracy

The first model included the sum-coded factors Group (Korean native vs. L2 Korean learners), Case (Accusative vs. Dative) and Order (Canonical vs. Scrambled), and their interactions. Participant was included as a random factor; item was not included due to non-convergence when added. No slope was included due to non-convergence of the model when added. The results are summarized in Table 1. 

Accuracy was lower overall for L2 learners than for Korean native controls (73% vs. 94%; β = −1.11, se = 0.20, z = −5.58, *p* < 0.001), for Accusative than Dative sentences (78% vs. 88%; β = −0.42, se = 0.13, z = −3.29, *p* < 0.001) and for Scrambled than Canonical word order (77% vs. 90%; β = 0.72, se = 0.13, z = 5.65, *p* < 0.00001). None of the interactions reached statistical significance (Case.sum × Order.sum, z < 1; Group.sum × Case.sum, z > 1; Group.sum × Case.sum × Order, z > 1), although there was a trend for the effect of Order to be modified by Group (β = −0.23, se = 0.13, z = −1.80, *p* < 0.07). Given the absence of interactions with Group, no further breakdown of the data was performed. 

##### Reading Times

Analyses were performed on the data for experimental sentences for which participants correctly answered the probe question. Data points exceeding the mean plus or minus two standard deviations were replaced by that number. The data were modeled independently for Accusative and Dative sentences due to differences in length of the nominal case markers and the design, which counterbalanced items across word order (Canonical and Scrambled), but only within sentence type (Accusative vs. Dative). To examine scrambling, we modeled the data for the first two ROI—N1 and N2—as independent regions, for both sentence types. The first model included the sum-coded factors Group (Koreans vs. Learners), Order (Canonical vs. Scrambled), ROI (N1 vs. N2), and their interactions. Participant and Item both included random intercepts and a random slope for Order. The final ROI (Verb) was modeled independently for both sentence types. While the models are reported for correct trials, it is noteworthy that the results did not differ from those conducted on all trials; indeed, for L2 learners, roughly 50% of trials were excluded due to error for scrambled accusatives. The data are summarized in Table 3 and the model outputs are presented in Table 4. 

##### Accusative Sentences

First Fixations

No effects were found in the analysis of first fixations. Given that there was neither an effect of Group nor any interaction with Group, no further models were performed.

First Pass Dwell Times

The analysis of first pass dwell times for N1 and N2 revealed effects of Group (β = −344.94, se = 64.89, t = −5.32, *p* < 0.001), of ROI (β = −153.94, se = 28.62, t = −5.38, *p* < 0.001), and their interaction (β = 112.21, se = 39.47, t = 2.84, *p* < 0.005). No other effects or interactions reached significance. The same, sum-coded model showed significant main effects of Group (β = −138.76, se = 33.25, t = −4.17, *p* < 0.001), of ROI (β = 40.52, se = 7.52, t = 5.39, *p* < 0.001) and their interaction (β = 28.62, se = 7.52, t = 3.81, *p* < 0.0002). Independent models were subsequently run on the data for the native Koreans and L2 learners, using the same model without the Group factor. 

For native Koreans, no effects were significant (Order (β = 39.25, se = 33.37, t = 1.18, ns); ROI (β = −33.58, se = 21.88, t = −1.53, ns); Order × ROI (β = 16.44, se = 31.86, t = 0.52, ns)). Despite numerical differences, native Koreans showed no variation in first pass reading times as a function of scrambling for accusative sentences when reading the first two nouns. No effects emerged at the Verb region (Order.sum = β = 9.18, SE = 12.24, t = 0.749, ns).

For L2 learners, the analysis of N1 and N2 revealed an effect of ROI (β = 68.98, SE = 13.29, t = 5.19, *p* < 0.001), but no effect of Order (β = −9.93, se = 39.71, t = −0.25, ns) or the interaction (β = 30.57, se = 53.71, t = 0.58, ns). L2 learners spent more time reading the first than the second noun, independent of Order (canonical or scrambled). No effects emerged at the Verb region (Order.sum: β = −22.50, se = 54.33, t = −0.41). It is of interest to note the same pattern of results emerged for analyses conducted on the entire data set; indeed, 47% of scrambled accusatives were excluded from the above analyses, which were conducted on correct trials only.

Total Dwell Times

The sum-coded model of total reading times, including both Korean natives and L2 learners, showed an effect of Group (β = 406.58, se = 42.89, t = 9.48, *p* < 0.001) Order (β = −61.809, se = 16.93, t = −3.65, *p* < 0.002) and their interaction (β = −50.59, se = 14.61, t = −3.46, *p* < 0.001). Independent models were subsequently run on the two groups. 

For Koreans, no effects emerged for N1 and N2 (Order (β = 35.56, se = 33.65, t = 1.06, ns); ROI (β = −27.37, se = 29.96, t = −0.91, ns); Order × ROI (β = −34.47, se = 42.76, t = −0.81, ns). Reading times did not differ as a function of scrambling during the processing of the first two nouns. Independent analyses of the verb region equally failed to reveal a significant effect of Order (β = 9.90, se = 23.02, t = 0.43, ns).

For the L2 learners, the sum coded model revealed an effect of Order (β = −108.91, se = 35.75, t = −3.046, *p* < 0.01), but no effect of ROI (β = 20.95, se = 28.64, t = 0.73, ns) nor their interaction (β = 24.24, se = 28.64, t = 0.40, ns). Total reading times were longer for scrambled than canonical word order, for both N1 and N2. No effects emerged at the Verb region (Order.sum: β = −22.50, se = 54.33, t = −0.41, ns). 

##### Dative Sentences 

First Fixations

The analysis of N1 and N2 revealed an effect of Group and its interaction with ROI (β = −14.16, se = 3.21, t = −4.41, *p* < 0.01). Independent models were run on the data for each Group. Native Koreans showed an effect of ROI, due to longer first fixation durations on N1 compared to N2, independent of Order. No other effects were significant. L2 learners showed no effects.

First Pass Dwell Times

The analysis of first pass dwell times at N1 and N2 revealed effects of Group (β = 206.33, se = 36.70, t = 5.62, *p* < 0.001), ROI (β = 59.41, se = 7.86, t = 7.56, *p* < 0.001), and the interactions between Group × ROI (β = 20.70, se = 7.86, t = 2.635, *p* < 0.01), Order × ROI (β =−58.58, se = 7.80, t = −7.51, *p* < 0.001), and the higher order interaction of Group × Order × ROI (β = −25.19, se = 7.80, t = −3.23, *p* < 0.001). Independent analyses were subsequently performed on the two groups. 

For Koreans, there was a trend for the effect of Order (β = 41.43, se = 21.28, t = 1.95, *p* < 0.06) and a significant interaction Order × ROI (β = −126.59, se = 26.21, t = −4.83, *p* < 0.001). For canonical word order, there was no effect of ROI (β = −14.41, se = 18.64, t = −0.77, ns), whereas there was a highly significant effect of ROI for scrambled word order (β = −140.95, se = 18.64, t = −7.56, *p* < 0.001). Korean native speakers showed longer reading times for the first than the second noun in scrambled dative sentences, whereas they showed no difference in reading times for the two nouns in canonical dative sentences. 

For L2 learners, a similar pattern emerged. There was an effect of Order (β = 169.98, se = 41.66, t = 4.08 *p* < 0.001) and a significant interaction of Order × ROI (β = −334.97, se = 58.07, t = −5.77, *p* < 0.001). For canonical word order, there was no effect of ROI (β = 7.292, se = 40.10, t = 0.18 ns), but for scrambled word order the effect of ROI was highly significant (β = −327.70, se = 42.00, t = −7.80, *p* < 0.001). L2 learners showed no difference in first pass dwell times for N1 and N2 for canonical dative sentences but were considerably slower to read the first (dative marked) noun than the second (nominative marked) noun for scrambled dative sentences. 

Total Dwell Times

The sum coded model, involving both groups, revealed a pattern similar to that found for first pass reading times. The model showed main effects of Group (β = −836.27, se = 124.79, t = −6.70 *p* < 0.001) and Order (β = 361.79, se = 59.12, t = 6.12, *p* <.001), and significant interactions between Group × Order (β = −203.50, se = 80.86, t = −2.517, *p* < 0.01), Order × ROI (β −351.83, se = 74.89, t = −4.70, *p* < 0.001) and the higher order interaction of Group × Order × ROI (β = 202.93, se = 103.36, t = 1.96, *p* < 0.05). Independent analyses were subsequently performed on the two groups. 

For native Koreans, there was an effect of Order (β = 157.48, se = 27.94, t = 5.64, *p* < 0.001) and a significant interaction between Order and ROI (β = −149.44, se = 36.99, t = − 4.04, *p* < 0.001). For canonical word order, there was no effect of ROI (β = −16.31, se = 26.17, t = −0.62, ns), whereas there was a highly significant effect of ROI for scrambled word order (β = −165.75, se = 26.47, t = −6.26, *p* < 0.001). This pattern mimics that found for first pass dwell times, with increased reading times for the first compared to the second noun specifically in scrambled dative sentences. 

For the L2 learners, a similar pattern emerged. There was a significant effect of Order (β = 355.64, se = 82.68, t = 4.30, *p* < 0.001) that was modified by the interaction with ROI (β = −352.16, se = 99.16, t = −3.55, *p* < 0.001). L2 learners took significantly longer to read the first than the second noun for scrambled sentences (β =−330.82, se = 72.21, t = −4.58, *p* < 0.001), but showed no difference in reading times for the two nouns for canonical word order (β = 21.19, se = 68.33, t = 0.31, ns). No reliable differences were found at the Verb region. 

#### 3.2.3. Regressions

As a final measure, we calculated whether readers made a regressive saccade to N1 and/or N2. Independent models were run on the data for Accusative and Dative structures. For Accusatives, the sum coded model involving both groups revealed an effect of Group and its interaction with ROI (β = −0.24, se = 0.10, z = −2.40, *p* < 0.02). Independent models on the two groups revealed an effect of ROI for both Koreans (β = 1.81, se = 0.14, z = 13.06, *p* < 0.001) and L2 learners (β = 1.24, se = 0.15, z = 8.19, *p* < 0.001) due to a greater probability to make a regressive saccade to N1 than to N2, independent of Order. The difference was larger, however, for Koreans. For Datives, the sum coded model involving both groups revealed only an effect of ROI (β = 1.30, se = 0.08, z = 14.69, *p* < 0.001), due to a higher probability to make a regressive saccade to N1 than N2, independent of Order. The data are summarized in Table 5.

### 3.3. Discussion

The present experiment examined the effect of scrambling on online sentence comprehension and reading times in native Koreans and L2 learners. In comparison to the results reported for Experiment 1, we found greater similarity in the pattern of results for the two groups as concerns comprehension, despite the lower accuracy and longer reading times in the L2 group. 

Concerning accuracy, both groups showed an overall decrease in sentence comprehension for scrambled sentences, as shown by the dip in accuracy for the probe questions following sentences with scrambled as compared to canonical word order. This was nonetheless substantially more apparent in the L2 group. Interestingly, the effect was true independent of sentence structure (accusative or dative) for both groups. In this vein, it is notable that L2 learners demonstrated higher accuracy in the present experiment, with written materials, than in Experiment 1, with auditory presentation, although for scrambled accusative structures, accuracy on the probe question was still at chance level. This result is reminiscent of that reported by Kim et al. (2018), who found that children who were heritage speakers of Korean performed better on a task requiring the processing of case when a prior context was provided than in the absence of such, but still did not perform above chance. 

The effect of scrambling on reading times differed for accusative and dative structures. Consider, first, the results for accusatives. For first pass reading times, no notable effects were found in either the native control or the L2 group; the only effect was an overall increase in reading times for N1 in the L2 group, independent of word order. Total reading times showed that while native controls continued to show no effect of scrambling, the L2 group showed longer reading times in scrambled sentences, for both nouns, compared to sentences with canonical word order. Thus, for accusative structures, the native control group showed no evidence of a cost of scrambling, whereas the L2 group did but only later in processing, during re-readings of the sentence. Moreover the effect was not specific to the first noun in the L2 group, i.e., they did not specifically hesitate at the first noun when marked for the accusative rather than the nominative, but took longer to read both nouns in scrambled than canonical word order. Finally, no effects were found at the verb, in either group, thus showing that participants were concentrating primarily on case marked nouns during processing. 

For dative structures, the effect of scrambling was apparent for both native controls and L2 learners, from the first pass through sentences to total reading times. In both groups, we found a significant increase in dwell times on the first noun compared to the second noun, specifically for scrambled sentences. For canonical structures, no differences were found for the reading times of the two nouns. This is important, as it rules out the possibility that the increased reading times for the first noun in scrambled dative sentences was merely a length effect. Indeed, the dative marker for animate nouns 에게 (eykey) is one syllable block longer than the nominative marker (이 (i) or 가 (ka) depending on the syllable structure of the noun). On average, across the set of our materials, the dative-marked noun was one syllable block longer than the nominative-marked noun. That this did not incur longer reading times of the second noun compared to the first noun in canonical (SOV) sentences allows us to conclude that the systematic increase in reading times for the dative marked noun in N1 position in scrambled sentences was not due to length. Rather, when encountering a dative marked noun in initial position, participants dwelled upon such to compute the sentence structure prior to moving on to the next noun in the sentence. This result is in line with numerous results from studies conducted in Japanese, showing that native speakers are slower to accept grammatically correct sentences when they are scrambled than in canonical order [52,53,54,63]. However, the evidence of a processing cost for scrambling in Japanese is more elusive in self-paced reading; Shibata et al. [57] found only marginal effects for native speakers, while none were reported in two other studies [55,58]. The lack of an effect in self-paced reading may nonetheless be tied to the measure itself, which may be too coarse to pick up on within clause scrambling costs [57]. 

The absence of an effect of scrambling on reading times was also reported by Mitsugi and MacWhinney [55] in a self-paced reading study conducted in Japanese that included a native control group and two L2 groups whose native languages differed as concerns the typological similarity to Japanese (English vs. Korean). The authors concluded that scrambling did not incur a processing cost and that the L2 learners’ native language did not play a predominant role. Two caveats are in order. First, the authors did not compare the groups statistically, but ran independent analyses on each. Second, the Korean learner group did in fact show an increase in reading time for the first noun in scrambled dative-accusative structures (N1dat-N2acc-N3nom) compared to scrambled accusatives (N1acc-N2nom-N3dat). Our results show that both Koreans and L2 learners showed increased reading times when the dative noun was fronted, but not for scrambled accusatives. Due to systematic differences in the number of syllable blocks for the dative and accusative case markers, we did not conduct a direct comparison of accusative and dative structures. Mitsugi and MacWhinney [55] hypothesized that the increased processing time for fronted dative-marked nouns in Japanese was due to the lower validity and cue strength, as concerns the assignment of syntactic roles of the dative compared to the accusative, rather than to costs associated with movement and restructuring. The pattern of results we obtained across the two structures does not lend support to the idea that the accusative has higher cue validity. Our L2 learners did not reliably use the accusative marker when fronted, i.e., in scrambled accusative sentences, whether auditory (Experiment 1, see also Frenck-Mestre et al. [8]) or written (Experiment 2). Moreover, comprehension was overall better for dative than accusative sentences across both presentation formats.

## 4. General Discussion

The present study used eye tracking to examine the online processing of nominal morphology for native Korean speakers and L2 learners in both spoken utterances (Experiment 1) and written sentences (Experiment 2). For both formats, materials were presented in either canonical (SOV) or scrambled (OSV) word order, and sentences contained either dative or monotransitive accusative verbs. Overall, our study provides a complex pattern of results, whereby presentation format, specific case marking and word order (scrambling) all played a role in the online processing of case. This was true for both native Koreans and L2 learners whose L1 was French. The pattern of online processing of case was more similar across native and L2 groups for written than auditory format, in line with several studies [5,6,8,9,11,15,35]. Our unique examination of processing across formats within the same study allows us to clearly demonstrate such. Moreover, the results of the present study provide a near perfect replication of results for the auditory processing of case, in both native Koreans and L2 learners (Frenck-Mestre et al. [8]).

For written sentences, we found that scrambling incurred a processing cost for both L2 learners and native speakers, but differentially as a function of case marking. These results corroborate and further our previous findings (Frenck-Mestre et al. [8]) and highlight the factors that affect the capacity to process case online in Korean. They also challenge certain findings concerning the impact of scrambling on native speakers’ performance in reading [55,58], see [57,63]. 

During auditory processing, native speakers rapidly exploited case information to compute the structure of the utterance as soon as enough information was available to rule out alternative interpretations. This was revealed by their looks to the correct image, starting from the second noun in the utterance and continuing through to the end of the utterance. In line with previous results [2,3,5,6], we did not find any effects prior to the onset of the second auditory noun. That is, despite having a full second to process the visual scene prior to the onset of the utterance, native speakers showed no preference for either image during the first auditory noun. In prior studies [2,3,5,6]), participants could only predict the target object once they had integrated information provided by the first elements. In our materials, although the first noun was overtly case marked, some ambiguity remained concerning its syntactic role up until the processing of the second noun whose case marking should have largely diminished any alternative interpretation (although, with the dative structures, some ambiguity could have remained up until the final verb). Moreover, because nominal case markers are affixed in Korean, this information comes into play relatively late for listeners to be able to launch a saccade to the correct image prior to the onset of the second noun [69]. Hence, one would not predict any effect prior to processing N2. Our native speakers clearly exploited case as soon as it became available and used it incrementally as the utterance unfolded to build a syntactic representation. This result replicates our previous study of the online auditory processing of case in Korean by native speakers [8] and is in line with numerous other studies of native speakers’ ability to process case [2,3,5,6,55,56,70,71]. 

For the group of L2 learners, the pattern of results found during auditory processing of case morphology largely replicates that found in previous online auditory studies, whereby these participants do not demonstrate an ability to exploit case to predict upcoming elements but wait until the end of the utterance, when all information is available, to compute structure and meaning. This has been found in Japanese [6], German [5], and Korean [8]. Moreover, when utterances contain scrambled elements, L2 participants show a strong initial reliance on word order over case morphology, as reported by Hopp [5] and Frenck-Mestre et al. [8]. This was mitigated, however, by the type of case marking, with a far stronger reliance on word order for accusative than dative structures. This effect of the specific type of nominal case marking replicates that found previously for adult L2 learners whose L1 was French, but not for those whose L1 was typologically more similar to Korean [8], and for heritage speakers of Korean, whose L1 was English [47]. 

By manipulating presentation format, we showed that the L2 learners were better able to exploit case information when allowed to process it at will. That is, in comparison to the results obtained for auditory materials, the L2 learners had higher accuracy for scrambled sentences during reading, most likely due to the possibility of either dwelling on elements that carried crucial syntactic information and/or re-reading them, which is not possible during auditory processing. This gain in comprehension was true for both dative and accusative sentences, but greater for the latter due to the extremely poor performance on scrambled accusatives during auditory processing. The increased capacity to process case morphology during reading is in line with the results of several eye movement and self-paced reading studies with L2 learners, showing that some can even perform at native like levels provided sufficient proficiency [9,11,15,35]. It is noteworthy that we found this gain for L2 learners who were far less proficient than those tested in previous studies. In addition, although our participants had the task of answering a probe question, said probe tapped comprehension and not the grammaticality of the sentence. Nonetheless, the probe question did necessitate the identification of the subject of the prior sentence, which may have directed participants’ attention to grammatical roles. Under such a hypothesis, the greater ability of the L2 learners to exploit case morphology during reading may partially be explained by their attention being drawn to this feature (Jackson and Bobb, 2009). 

In relation to the potential cost of scrambling, our results provide distinct answers depending on the presentation format, case marking and participant group. During auditory processing, the native Korean group showed no processing cost linked to scrambling, either in terms of accuracy or the immediate use of case to determine the structure of the utterance (cf also [8]). In contrast, the L2 group showed a definite cost of scrambling during auditory processing, which was impacted by the specific structure, with scrambled monotransitive accusatives posing greater disruption than scrambled datives, which replicates previous findings [8,47]. The impact of scrambling was more pronounced during reading, with both the native and L2 group demonstrating significantly longer reading times at the first compared to the second noun for scrambled dative sentences. For accusative structures, only the L2 group showed an increase in reading time for scrambled sentences and such was true for both the first and second noun. Hence, during reading, scrambled datives produced a specific processing cost at NP1 for all participants, but accusatives only produced a general slowing and only in the L2 group. 

The pattern of results we obtained during reading suggests that for either reasons of syntactic complexity or comparative frequency, scrambling is costly, even for native speakers. This result is in line with several studies of Japanese using offline measures, which showed that grammaticality judgements were slowed for scrambled sentences, albeit for far more complex ones than those used herein [52,53,54,63]. This contrasts with previous online studies that used self-paced reading and showed either minimal or no effects of scrambling [55,57,58]. By recording eye movements, as opposed to the coarser measure that is provided by manual responses in self-paced reading, we were able to evidence the transient nature of the difficulty associated with scrambling. However, as highlighted by Shibata et al. [57] and as underscored by the difference in patterns we obtained for accusative and dative structures, within clause scrambling may produce effects that go undetected when restructuring is relatively easy, as was apparently the case for monotransitive accusative sentences. 

Last, we can note that across both auditory and written processing, we found distinct patterns of processing as a function of case marking. Dative structures resisted scrambling better than accusative structures for both auditory and written format for L2 learners as revealed by both accuracy data and reading times (cf. Kim et al. [47], for auditory processing, in heritage speakers). During reading, both native and L2 learners showed a cost of scrambling for dative structures, whereas only the L2 learners showed an increase in reading time for scrambled accusatives and such was not specific to the first, scrambled noun. The differences in processing for these two structures may be attributed to numerous factors. The dative case in Korean differs from the accusative both semantically and syntactically [20,39,42,43,44,46]. In addition, the dative and accusative differ as concerns their phonetic form, which may well affect their saliency during auditory processing, although this has been claimed more in the context of non-native than native processing [8,47]. Dative case is also more reliable than accusative in that the dative is rarely omitted whereas accusative may be dropped depending upon discourse constraints [43,44,59]. In addition, the group of second language learners in the present study, were native speakers of French, which has a distinctive morphological marker for the dative but not accusative or nominative. This may have contributed to these learners acquiring the dative earlier in their L2 trajectory [62]. Any or all of these factors may have played a role in the differential effects we found for monotransitive accusative and dative structures. This difference in processing as a function of the specific type of case marking has been attested in several studies and should hence be taken into account in future research [8,47]. 

In conclusion, the present set of experiments provides clear evidence of the incremental nature of syntactic processing involving the immediate extraction of case to determine sentence structure for native speakers, but a strong reliance on word order over grammatical case for L2 learners. For the latter group, the apparent inability to compute nominal case morphology during online processing was more pronounced during listening than reading, in line with several studies of L2 processing. Importantly, the effect of scrambling in L2 processing differed according to the specific case marking, a factor that has not systematically been explored in previous work. This adds to previous native data in both auditory and written processing [8,70,71]. Lastly, for both native speakers and L2 learners, we provide evidence of a transient cost of scrambling during reading, even for syntactically simple declarative sentences. 

## Figures and Tables

**Figure 1 brainsci-12-01230-f001:**
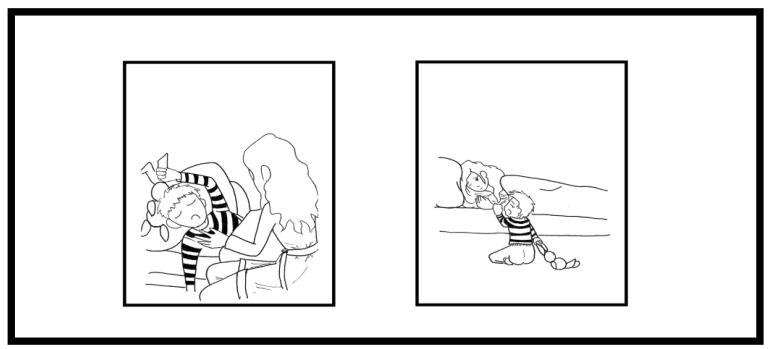
Example of line drawings presented simultaneously with the auditory sentence in Experiment 1.

**Figure 2 brainsci-12-01230-f002:**
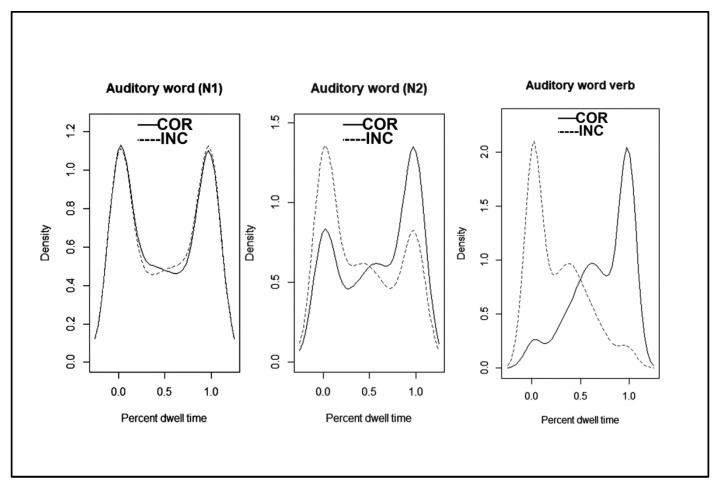
Density plots for Korean participants for all trials showing the percentage of dwell times in the correct (COR) and incorrect (INC) image for the duration of each auditory word.

**Figure 3 brainsci-12-01230-f003:**
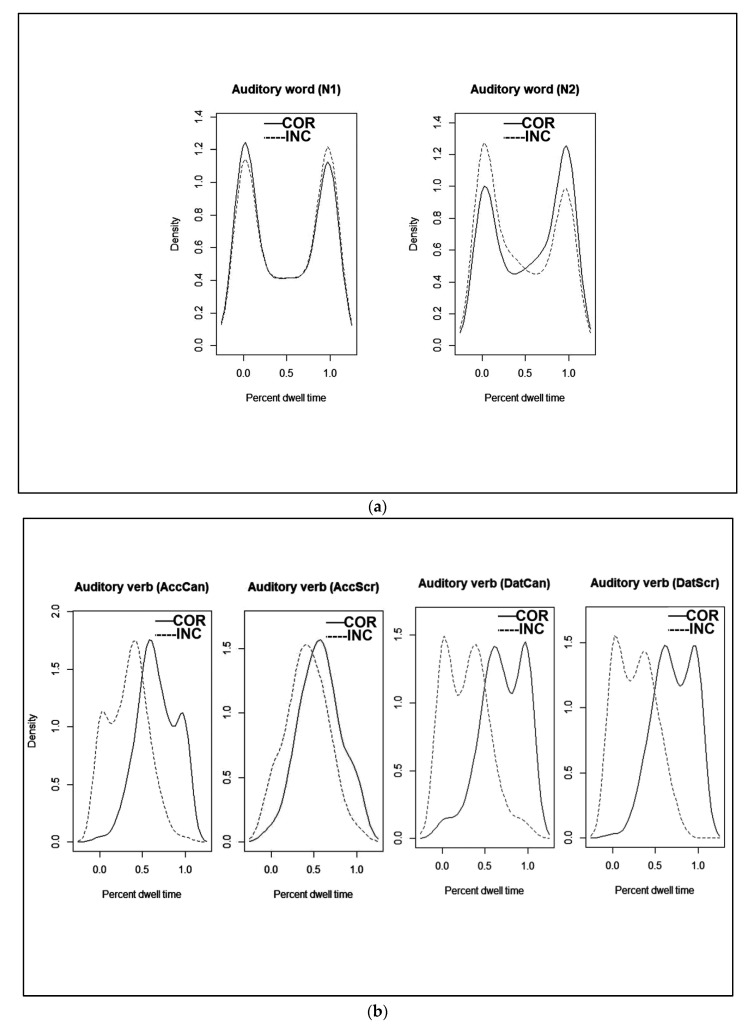
(**a**). Density plots for L2 participants for all trials for the percentage of dwell times in the correct (COR) and incorrect (INC) image for the duration of N1 and N2. (**b**). Density plots for L2 participants for all trials showing the percentage of dwell times in the correct (COR) and incorrect (INC) image for the duration the verb as a function of Case (Dat = dative, Acc = accusative) and Order (Can = canonical, Scr = scrambled).

**Table 1 brainsci-12-01230-t001:** Accuracy rates in Experiment 1 (auditory processing) and Experiment 2 (reading) for each Group (Koreans vs. L2 learners) as a function of Case (accusative vs. dative) and Order (CAN = canonical vs. SCR = scrambled).

	Experiment 1				Experiment 2			
	Accusative		Dative		Accusative		Dative	
	CAN	SCR	CAN	SCR	CAN	SCR	CAN	SCR
Koreans	98(16)	98(14)	94(24)	95(22)	98(16)	86(35)	99(11)	93(36)
L2 learners	87(33)	29(46)	77(42)	63(48)	75(43)	54(46)	87(33)	75(44)

**Table 2 brainsci-12-01230-t002:** Percentage of dwell times (SD in parentheses) for each Group (Koreans vs. L2 learners) in each image (COR = correct, INC = incorrect) as a function of Case (accusative vs. dative), Order (canonical vs. scrambled), and ROI (N1, N2, (N3), VB) for correct trials and all trials.

	L2 Learners				Koreans	
Correct Trials						
Accusative	N1	N2	VB	N1	N2	VB
**Canonical**						
COR	50.6(43)	55.2(42)	66.9(22)	51.4(43)	58.2(42)	72.5(31)
INC	49.2(43)	44.7(42)	33.0(22)	48.5(43)	40.9(42)	27.5(31)
**Scrambled**						
COR	42.1(45)	57.7(44)	57.5(24)	53.8(43)	59.7(44)	73.7(28)
INC	56.7(45)	41.3(44)	41.6(24)	46.2(43)	40.1(44)	26.3(28)
**Dative**						
**Canonical**						
COR	50.6(44)	52.3(40)	70.9(25)	49.6(40)	57.6(36)	69.5(32)
INC	49.3(44)	47.7(40)	28.9(25)	50.4(40)	42.4(36)	28.2(31)
**Scrambled**						
COR	44.4(41)	53.4(40)	72.1(22)	41.9(40)	57.4(38)	74.9(29)
INC	55.6(41)	44.9(40)	27.2(22)	57.7(41)	42.5(38)	22.0(25)
**All Trials**						
**Accusative**	**N1**	**N2**	**VB**	**N1**	**N2**	**VB**
**Canonical**						
COR	52.3(43)	53.8(42)	62.9(25)	50.8(43)	58.2(42)	71.5(32)
INC	47.9(43)	46.2(42)	37.0(25)	48.5(43)	41.0(42)	27.8(31)
**Scrambled**						
COR	45.9(44)	50.7(44)	43.9(23)	53.4(43)	59.9(44)	72.7(28)
INC	53.6(44)	48.8(44)	55.6(23)	46.6(43)	40.0(44)	26.9(28)
**Dative**						
**Canonical**						
COR	51.9(44)	54.4(39)	62.6(28)	48.6(41)	56.3(36)	70.6(31)
INC	47.8(44)	45.6(39)	37.1(28)	51.4(41)	43.7(36)	29.4(31)
**Scrambled**						
COR	48.2(41)	50.2(40)	59.4(28)	42.2(41)	56.9(38)	74.8(28)
INC	51.8(41)	48.9(40)	40.1(27)	57.4(41)	43.1(38)	25.2(28)

**Table 3 brainsci-12-01230-t003:** Mean reading times in ms and SD (in parentheses) as a function of Case (accusative vs. dative), Order (canonical vs. scrambled), ROI (N1, N2, (N3), VB), and Group (Koreans vs. L2 learners) for correct trials and all trials.

	L2 Learners				Koreans			
Accusatives: Correct Trials								
**Canonical**	N1	N2		VB	N1	N2		VB
First fixation	282(128)	267(101)		329(166)	258(99)	238(127)		176(102)
First pass gaze	731(338)	573(313)		575(317)	364(150)	360(220)		220(129)
Total	1339(653)	1241(596)		824(534)	601(247)	597(285)		298(228)
**Scrambled**								
First fixation	275(104)	278(119)		305(115)	264(81)	242(127)		164(82)
First pass gaze	741(348)	617(314)		546(320)	362(146)	416(265)		210(132)
Total	1573(701)	1571(625)		987(520)	634(266)	604(219)		293(266)
**Accusatives: All Trials**								
**Canonical**	N1	N2		VB	N1	N2		VB
First fixation	286(127)	273(109)		336(158)	256(98)	236(125)		175(101)
First pass gaze	749(340)	598(321)		617(340)	368(152)	365(220)		221(129)
Total	1360(665)	1303(606)		903(573)	601(247)	597(285)		298(228)
**Scrambled**								
First fixation	267(103)	284(121)		321(134)	262(77)	243(123)		160(79)
First pass gaze	753(335)	617(311)		550(311)	363(147)	420(269)		203(127)
Total	1525(681)	1458(620)		930(510)	628(245)	590(226)		275(250)
**Datives: Correct Trials**								
**Canonical**	N1	N2	N3	VB	N1	N2	N3	VB
First fixation	274(127)	287(126)	327(143)	345(160)	258(91)	225(106)	213(87)	195(87)
First pass gaze	762(402)	770(451)	697(320)	669(339)	371(147)	383(208)	250(87)	211(85)
Total	1412(698	1438(685)	1001(515)	923(511)	577(213)	585(237)	365(211)	270(187)
**Scrambled**								
First fixation	248(89)	274(121)	316(123)	320(135)	256(72)	216(88)	218(89)	180(93)
First pass gaze	938(473)	614(320)	655(306)	703(372)	428(154)	294(135)	250(102)	200(91)
Total	1775(890)	1454(653)	1113(574)	970(568)	734(257)	581(258)	381(225)	297(233)
**Datives: All Trials**								
**Canonical**	N1	N2	N3	VB	N1	N2	N3	VB
First fixation	275(124)	284(120)	337(151)	343(156)	257(91)	225(106)	213(87)	195(87)
First pass gaze	802(409)	797(450)	697(311)	682(350)	369(148)	382(207)	250(87)	211(85)
Total	1467(694)	1478(681)	1052(547)	939(537)	573(215)	583(237)	365(211)	270(187)
**Scrambled**								
First fixation	250(110)	272(118)	320(141)	329(142)	257(70)	212(87)	216(88)	178(92)
First pass gaze	943(463)	629(325)	668(319)	727(390)	425(152)	296(137)	251(102)	198(91)
Total	1748(856)	1414(632)	1112(551)	9990(581	728(254)	576(260)	381(220)	291(231)

**Table 4 brainsci-12-01230-t004:** Fixed effects for the models including both groups and for each group, for the first and second noun and each of the three reading time measures, for accusative and dative structures. All significant effects are indicated by an asterisk.

	First Fixations			First Pass Dwell Times			Total Reading Times		
**Accusatives**									
**Fixed effects**	Estimate	*SE*	*t*	Estimate	*SE*	*t*	Estimate	*SE*	*t*
**Koreans + Learners** (Intercept)	262.93	7.01	37.50 **	546.49	33.88	16.13 *	1031.65	43.95	23.48 *
GROUP.sum1	12.06	6.62	1.82	138.76	33.25	4.17 *	406.58	42.89	9.48 *
ORDER.sum1	−1.72	4.48	−0.38	−9.58	8.49	−1.13	−61.81	16.93	−3.65 *
ROI.sum1	6.81	3.78	1.80	40.52	7.52	5.39 *	22.00	14.46	1.52
ORDER.sum1:ROI.sum1	1.53	3.74	0.41	8.39	7.39	1.14	6.66	14.25	0.47
GROUP.sum1:ROI.sum1	−3.93	3.78	−1.04	28.62	7.52	3.81 *	−1.16	14.46	−0.08
GROUP.sum1:ORDER.sum1	1.79	3.84	0.47	5.67	8.32	0.68	−50.59	14.61	−3.46 *
GROUP.sum1:ORDER.sum1:ROI.sum1	2.80	3.74	0.75	−0.57	7.40	−0.08	17.48	14.25	1.23
**Koreans** (Intercept)	*NA*	*NA*	*NA*	408.93	34.45	11.87*	−19.05	14.77	−1.29
ORDER.sum1	*NA*	*NA*	*NA*	−23.73	13.99	−1.70	21.81	13.46	1.62
ROI.sum1	*NA*	*NA*	*NA*	12.68	8.14	1.56	5.05	13.03	0.39
ORDER.sum1:ROI.sum1	*NA*	*NA*	*NA*	4.11	7.96	0.52	−19.05	14.77	−1.29
**Learners** (Intercept)	*NA*	*NA*	*NA*	687.12	57.53	11.94 *	1439.00	88.60	16.24 *
ORDER.sum1	*NA*	*NA*	*NA*	−2.68	14.74	−0.18	−108.91	35.75	−3.05 *
ROI.sum1	*NA*	*NA*	*NA*	68.98	13.29	5.19*	20.95	28.64	0.731
ORDER.sum1:ROI.sum1	*NA*	*NA*	*NA*	7.64	13.29	0.5	24.24	28.64	0.846
**Datives** **Fixed Effects**									
**Koreans + Learners** (Intercept)	255.35	5.84	43.70 *	585.36	38.28	15.22 *	1088.01	65.30	16.66 *
GROUP.sum1	16.09	5.51	2.92 *	206.33	36.70	5.62 *	453.39	64.47	7.03 *
ORDER.sum1	7.29	4.10	1.78	2.52	8.44	0.30	−67.43	16.36	−4.12 *
ROI.sum1	4.55	3.21	1.42	59.41	7.85	7.56 *	61.74	13.00	4.75 *
ORDER.sum1:ROI.sum1	0.81	3.20	0.25	58.58	7.80	−7.51 *	−62.59	12.92	−4.84 *
GROUP.sum1:ROI.sum1	−14.16	3.21	−4.41 *	20.70	7.85	2.63 *	15.62	13.00	1.20
GROUP.sum1:ORDER.sum1	3.93	3.61	1.09	−5.38	7.96	−0.68	−25.51	15.19	−1.68
GROUP.sum1:ORDER.sum1:ROI.sum1	2.32	3.20	0.72	−25.19	07.80	−3.23 *	−25.37	12.92	−1.96 *
**Koreans** (Intercept)	238.92	7.59	31.49 *	380.17	19.57	19.43 *	634.60	31.52	20.13 *
ORDER.sum1	2.77	3.97	0.70	10.87	8.60	1.26	−41.38	9.92	−4.17 *
ROI.sum1	18.72	3.82	4.89 *	38.81	6.63	5.86 *	45.52	9.36	4.86 *
ORDER.sum1:ROI.sum1	−1.77	3.79	0.47	−31.78	6.55	−4.85 *	−37.36	9.25	−4.04 *
**Learners** (Intercept)	270.73	9.19	29.46	789.97	73.98	10.67 *	1541.27	126.68	12.17 *
ORDER.sum1	11.91	7.67	1.55	−2.79	19.20	0.14	−89.78	33.08	−2.71 *
ROI.sum1	−9.64	5.22	1.85	80.11	14.30	5.60	77.44	24.79	3.12 *
ORDER.sum1:ROI.sum1	3.07	5.22	0.59	−83.65	14.30	−5.85	−88.04	24.79	−3.55 *

R model: (lmer(ADJ-DWT~(1 + ORDER.sum|Participant) + (1 + ORDER.sum|Item) + GROUP.sum:ROI.sum:ORDER.sum).

**Table 5 brainsci-12-01230-t005:** Likelihood to make a regressive saccade (SD in parentheses) to N1 and N2 during reading for each Group (Koreans vs. L2 learners) as a function of Case (accusative vs. dative) and Order (canonical vs. scrambled).

	Accusative				Dative			
	**Canonical**		**Scrambled**		**Canonical**		**Scrambled**	
	**N1**	**N2**	**N1**	**N2**	**N1**	**N2**	**N1**	**N2**
Koreans	82(39)	19(39)	80(40)	15(36)	84(37)	32(47)	81(39)	37(49)
Learners	85(36)	41(49)	93(26)	48(50)	87(33)	37(48)	91(28)	46(50)

## Data Availability

The data supporting the reported results are stored at the Laboratoire Parole et Langage UMR 7309 CNRS-Aix- Marseille University and are accessible upon request.

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
