# Peer review of "The Online Processing of Korean Case by Native Korean Speakers and Second Language Learners as Revealed by Eye Movements"

_brainsci, 2022, doi:10.3390/brainsci12091230_

Round 1

Reviewer 1 Report

Summary

The main question here is if differences in morphosyntactic processing for L2 learners (compared to L1 speakers) are due to an underuse of syntactic information or limited processing capacities, such as proficiency level, working memory capacity, and task demands. The paper accomplishes this aim via two experiments with L1 Korean speakers and intermediate L2 learners of Korean whose L1 was French.

Both experiments utilized a visual world paradigm to compare processing time for case marking and word order using eye tracking for the two groups. Experiment 1 used an auditory paradigm and Experiment 2 adapted the same materials to a reading experiment. The results of Experiment 1 revealed that during auditory processing, native Korean speakers showed incremental processing based on case information, with no effect of scrambling or specific case marking; however, L2 speakers showed no evidence of predictive processing and was negatively impacted by scrambling, especially for the accusative. The results of Experiment 2 revealed that during reading, both the native Korean speakers and the L2 group showed a cost of scrambling on first pass reading times, specifically for the dative and L2 learners showed better comprehension for scrambled dative than accusative structures across formats.

The authors present an overall conclusion that the modality of presentation is significant, that the specific type of nominal case plays a role, and that L2 learners show a pattern of processing more similar to that of native speakers during reading than auditory processing.

General Comments

The paper is well written, the methods are detailed, the motivation for the two experiments is explained logically, and the results are presented clear. The authors do an excellent job presenting the data and interpreting the results in a meaningful and logical way. There are some minor issues, including missing figure and table titles and some misspellings/typos, but overall, this is a well-written paper and I support its publication following a few minor revisions.

Minor revisions needed:

Page 1, line 41: commas are needed after each citation in the following sentence. “…including Korean (Frenck-Mestre et al., 2018), German (Hopp, 2015), and Japanese 41 (Mitsugi & MacWhinney, 2016).

Page 13, line 528: the word “included” is written twice.

Page 16, line 626: e.g. is written as eg. There should be a period after e and after g.

In the write up for Experiment 2, the term self-paced is used several times. It is currently written as “self paced”, but I believe the correct way to write this is self-paced.

Page 21, Table 4. The title for this table is missing.

Page 24, line 905, the word Accusative is misspelled.

Page 24, line 919: The word accuracy is misspelled.

Page 24, 930: The word heritage is misspelled.

Page 25, line 950: There should be a comma after the phrase, “For canonical structures, no differences…”

Page 25, line 996: The word tracking is misspelled.

Pages 13-14: Figures 2, 3a, and 3b should have titles and should include the group (i.e., Figure 2 shows density plots for Korean participants and Figures 3a and b show density plots for L2 learners). Also, why are additional plots shown for the L2 learners (i.e., AccCan, AccScr,DatCan, and DatScr) but not for the L1 native speakers in Figure 2? If there is a rationale for this difference, it needs to be explained.

Author Response

see as attachments

Reviewer 2 Report

It was a well-done job.

Author Response

Thank you for your comments

Reviewer 3 Report

This is a well executed study that I think is nearly ready for publication.  I feel that it could do with a quick copy editing though (I've noted some of the small errors in the pdf). In addition, I would like the authors to further clarify these issues in their revision (especially to assist with future researchers' replications) for clarity in procedures as well as the focuses of the experiments.

Please provide the detailed ethics information including approval number as well as the full name of the ethics review board(s).

For the abstract, could you please provide a sentence or two of background information in the abstract to explain the significance of this research topic. 

In the introduction I have pointed out a place where it might be helpful to directly state why you are conducting this investigation (to drive home for your claim for significance). 

You  have some predictions but I am wondering whether you considered writing them out as hypotheses? Doing this and then directly stating in the results whether the hypotheses were confirmed would be useful to readers. 

Please provide adequate information on the eye tracking equipment. I do not see the model number or some detailed settings indicated in the paper. These would be useful for replication purposes. 

Do  you plan to make your stimuli available to other researchers for replication purposes? That would be useful. 

Please provide more information on the proficiency of the Korean L2 learners. I am not sure if instruction hours is enough to understand their proficiency. 

Please justify which eye movement measures were collected and why those were the most useful for addressing your hypotheses. 

Please see the pdf for some other minor issues. 

Author Response

see as attachments
